**Subject Category:**
Biology (whole organism)

behaviour/evolution/ecology

cooperation, religion, evolution, reputation, punishment, China

**Author for correspondence:**
Ruth Mace
e-mail: r.mace@ucl.ac.uk

†These authors contributed equally to this study.

# Large-scale cooperation driven by reputation, not fear of divine punishment

Erhao Ge[1,†], Yuan Chen[1,†], Jiajia Wu[1] and Ruth Mace[1,2]

[1]State Key Laboratory of Grassland and Agro-ecosystems, School of Life Sciences, Lanzhou University, 222 Tianshui South Rd, Lanzhou, Gansu Province 730000, People's Republic of China
[2]Department of Anthropology, University College London, 14 Taviton Street, London WC1H 0BW, UK

EG, 0000-0001-6867-4595; RM, 0000-0002-6137-7739

Reputational considerations favour cooperation and thus we expect less cooperation in larger communities where people are less well known to each other. Some argue that institutions are, therefore, necessary to coordinate large-scale cooperation, including moralizing religions that promote cooperation through the fear of divine punishment. Here, we use community size as a proxy for reputational concerns, and test whether people in small, stable communities are more cooperative than people in large, less stable communities in both religious and non-religious contexts. We conducted a donation game on a large naturalistic sample of 501 people in 17 communities, with varying religions or none, ranging from small villages to large cities in northwestern China. We found that more money was donated by those in small, stable communities, where reputation should be more salient. Religious practice was also associated with higher donations, but fear of divine punishment was not. In a second game on the same sample, decisions were private, giving donors the opportunity to cheat. We found that donors to religious institutions were not less likely to cheat, and community size was not important in this game. Results from the donation game suggest donations to both religious and non-religious institutions are being motivated by reputational considerations, and results from both games suggest fear of divine punishment is not important. This chimes with other studies suggesting social benefits rather than fear of punishment may be the more salient motive for cooperative behaviour in real-world settings.

## 1. Introduction

Why humans help strangers in large-scale society is a puzzle. Many theoretical models and behavioural experiments suggest

cooperation takes place through indirect reciprocity, which is the result of bystanders' evaluation of your behaviour [1–3], or reputation-based partner choice, in which individuals reap benefit in cooperative reputation through access to more cooperative partners [4,5]. But theoretical models predict the evolution of cooperation based on reputational concerns becomes difficult as the group size increases [6–8]. Some evidence suggests that observing other group members or communicating reputation between individuals becomes more difficult as group size becomes larger [9–11]. Several experimental studies have found an association between group/population size and cooperation in the laboratory and field: people in larger groups may be more cooperative [12–14] or less cooperative [15], or that relationship may be nonlinear [16–18], but it should be noted that group size or population size may not be necessarily associated with community size, to our knowledge the role of residential community size has not been directly tested.

Stability may also enhance cooperation. One experiment found observability dramatically increased cooperation among those who own their homes/apartments relative to those who rent, who are likely to be more transient [19]. A study of hunter–gatherers in the Philippines showed that stable camps (with fewer changes in membership over time) were associated with greater reciprocal sharing [20].

Some argue that institutions are necessary to coordinate cooperation in large-scale societies [21–23]. Religion is such an institution that may be a cultural facilitator and play a crucial role in maintaining and raising levels of cooperation of large-scale societies [24–26]. Multiple facets of religion have emphasized its role as drivers of prosociality [27–31], but empirical findings on the relationship between religiosity and prosocial behaviour are mixed [32–35]. Some maintain that fear of monitoring or punishment from supernatural agents predisposes us to conduct costly prosocial behaviour, and that sanctified rituals, which serve to internalize religious commitment, discourage individuals from violating morality and motivate cooperation [36–47]. Others deem that religious beliefs and practice facilitate human prosocial tendencies due to reputational concerns; individuals who worship more frequently and carry out costly religious acts are seen as having a suite of reputational qualities by their peers [48,49]. Costly religious practice or rituals are likely to strengthen the trust between in-group members to reinforce cooperation and resist defection from free riders. There is also good empirical evidence that people seek to improve their prestige because of the many reputational benefits that come with a good social reputation (e.g. gaining access to resources [50], more potential helpers to draw upon [51], receiving help during illness, etc [52]). According to this view, religious prosocial behaviour may be mediated through fear of punishment from co-religionists in society, not from divine entities. It is possible that the two mechanisms outlined above are not mutually exclusive but complementary [53].

We established an experimental laboratory-in-field set-up by members of real residential communities. To distinguish between reputational and religious influences on prosociality, we had participants engage in two games based on the opportunity to donate to real-world religious and non-religious institutions: one in which participants' decisions to donate were directly observed (which we call the 'free donation game') and one in which their decision to donate was masked by a 'random' event, which was only observable to the participant, thus allowing subjects to deceptively opt out of donating (which we call the 'dice allocation game') (figure 1). These two games allowed us to determine whether the effects of religiosity on cooperation are driven by reputation concerns, fear of divine punishment, or both.

We conducted our two games on a single same sample from 17 different communities, large and small, comprising Muslims, Buddhists and atheists. Because all sites are within the same country, thus sharing much in terms of social and political context, we avoid some of the many potential confounds introduced by attempting to correlate across different countries (see electronic supplementary material, figure S1 for study locations). Both Buddhism and Islam are moralizing religions, which promote beliefs that future destiny is related to moral behaviour in one's current life by judgemental and moralistic divine punishment. We combined games with a detailed ethnographic survey to assess whether public religious practice and belief in divine punishment/reward are associated with endowments to institutions, as well as examining the role of community size.

We assume that reputational considerations will be more salient in small communities, which are also more stable as residents have lived there for a much longer time, and thus have a greater opportunity to get to know each other. In the 'free donation' game, we predict that if donations are driven by reputational concerns donations will be larger in small communities and by individuals engaging in more public religious practice (H1). We expect that belief in divine punishment will be associated with larger donations if donations are motivated by fear of supernatural monitoring (H2). In the 'dice allocation' game, cheating will be less when donating to religious institutions if donation behaviour is motivated by fear of divine punishment (H3). In addition, we predict community size is not important in the dice game as reputational considerations are removed (H4).

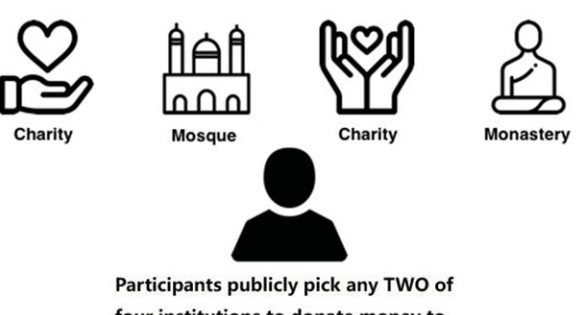

**Figure 1.** Experimental procedures. Participants were randomly allocated the order of game-playing (i.e. whether they play the Free Donation game or Dice Allocation game first).

## 2. Methods

### 2.1. Participants

A total of 501 people participated from 17 locations among large and small communities across populations from Muslim, Buddhism and some mixed communities in Gansu Province and Qinghai

Province, China. We use population size and geographical location to define two groups of communities, the small versus large communities, with 'small' indicating communities in villages and small towns (most of the small communities have a population of several hundred), and 'large' are cities (all these large communities have a population more than one hundred thousand). Six locations are defined as in large communities and eleven as in small communities (details are provided in electronic supplementary material, tables S1 and S2). The mean age of participants was $42.5 \pm 17.4$ years, and 60.9% of participants were male. Data collection was carried out from December 2016 to April 2017 by recruiting participants at random in various locations.

## 2.2. Measures and procedure

First, each participant was informed that they could publicly pick any two of four institutions to donate money to, consisting of two secular charities and two religious institutions. The two charities we offered were 'Mother's cellar' and 'Hope projects', which are well known in northwest China. Religious institutions were local mosques and Buddhist monasteries, selected from the nearest religious institutions (or the county's most famous religious institutions if no institution was nearby for one or other religion). All participants undertook two games, two rounds for each, donating to each institution they selected. All game players were anonymous to the experimenters (no names were recorded) and participation was voluntary (see electronic supplementary material, table S3 for the overall distributions of participants' choice of institutions).

### 2.2.1. Free donation game

Participants played two rounds of a free donation game, each round donating to one of the two recipient institutions they picked. We provided the player with 10 RMB for each round of the game (locally called 10 yuan). Participants were free to keep it all for themselves, or they could choose to allocate endowment in 2 RMB increments to the charity/religious institutions. And the whole game process was observable to others. The number of onlookers, any passerby and subsequent or previous participant who was around and watching, was recorded at the time of the participants were making their donation decisions. We experimenters were not included as onlookers.

### 2.2.2. Dice allocation game

Participants played two rounds of a dice allocation game, with 10 RMB each round, and each round concerning donations to each of the recipient institutions they picked (as in the free donation game). All the dices were pretested for bias (see electronic supplementary material, table S10). Participants were instructed to roll a dice placed in an opaque, covered cup twice, but to report the first or the second roll, randomized for order effects (following the procedures used by Gächter *et al.* [54]). The dice were thus unobservable by anyone except the participant. Participants were paid according to the number they reported, and the rest given to the corresponding institution. Dice throws claimed as '1,2,3,4,5', participants earned '2,4,6,8,10' RMB, while reported '6' participants earned nothing. Participants could cheat in reporting these dice throws as they were unverifiable. If all participants reported honestly, then each number should appear with an equal probability of 1/6.

Participants were afterwards asked to complete a brief questionnaire on demographic and socioeconomic status information. For measuring the religiosity, we asked about the importance of religion, how frequently they undertook religious practices and asked religious organizations for help, whether they had religious donation behaviour before, how much they believed in divine punishments and reward, and in the existence of a range of supernatural things, along with the geographical distance between experiment location and the religious institution they chose. The participants got their earning from the whole experiment when they finished the questionnaire and we allocated what participants donated to the institutions after we finished all the experiments (see electronic supplementary material, table S5 for further details).

## 2.3. Statistical analysis

We used principal component analysis (PCA) to analyse variables measuring participants' religiosity, such as 'Importance of religion', 'Religious donation', 'Religious Activities', 'Appeal to religious institutions', 'Belief in God punishment', 'Belief in God rewards', 'Belief in invisible things', 'Belief in

**Table 1.** Component loadings of principle component analyses for nine religiosity items. Results are identical for correlation matrix. All items are standardized by using z-score normalization, mapping on to three latent variables. Loadings more than 0.5 are in bold.

|  | religious practice (PC1) | divine punishment/ reward (PC2) | other supernatural beliefs (PC3) |
|---|---|---|---|
| importance of religion | **0.66** | 0.41 | 0.02 |
| religious donation | **0.62** | 0.15 | −0.07 |
| religious activities | **0.72** | 0.2 | 0.15 |
| appeal to religious institution | **0.75** | 0.12 | 0.18 |
| belief in God punishment | 0.15 | **0.86** | 0.15 |
| belief in God rewards | 0.14 | **0.87** | 0 |
| belief in invisible things | 0.43 | 0.19 | **0.59** |
| belief in supernatural power | 0.06 | 0.01 | **0.9** |
| religious institution distance | **0.64** | −0.05 | 0.05 |
| **eigenvalue** | 3.30 | 1.19 | 1.08 |
| **SS loading** | 2.55 | 1.79 | 1.24 |
| *proportion of variance* | *0.28* | *0.20* | *0.14* |
| *cumulative variance* | *0.28* | *0.48* | *0.62* |

supernatural power', 'Religious institution distance', and they were synthesized into three component factors (table 1). In the free donation game, 65% of donations were donating all (100%) of the RMB, so we used all endowment to the institution or not as the response variable in the analysis. We estimated generalized linear mixed models with a binomial distribution and a logit link function to analyse the donation game outcomes at the individual level. The full model encompassed socio-demographics such as gender, age, offspring, level of education, economic status and onlookers as controls, and three components of religiosity and institution choice as individual-level predictors, and community size as community-level predictor. Survey site was treated as random effect to control for the random variation in different sites. To affirm the robustness of our analyses, we also estimated a hurdle model predicting the magnitude of continuous values donated [55].

We selected the optimum model based on the Akaike information criterion (AIC) [56] to identify whether individual-level religiosity and/or community size best predicts donations in this free donation game. Parameter estimates were obtained by weighted-support model averaging from the entire set of candidate models.

In the dice allocation game, we did a general linear regression using a permutation test to identify determinants of dishonesty across different communities. We measured cheating at the community level by payoff distribution with reference to the expected flat distribution. The full model included proportion female, average economic status and average level of education of participants as controls, and community size and mean scores of three components of religiosity at community level as predictors. We used model selection to identify whether community-level religiosity and/or community size best predicts mean payoff. We selected the optimum model based on the second-order Akaike information criterion (AICc) values. We also estimated alternative measures of community-level dishonesty (see electronic supplementary material, table S13) controlling for sociodemographic respectively as a function of community size and religiosity for confirming the robustness of analyses.

We carried out all the statistical analysis in R v. 3.4.1 [57] with lme4, AICcmodavg and lmPerm Packages [58,59].

## 3. Results

Participants in small communities have lived there for much longer than those in big communities across our study sites, which we assume is greatly increasing the likelihood that onlookers are familiar with, or previously interact with, participants in small communities (figure 2, with electronic supplementary

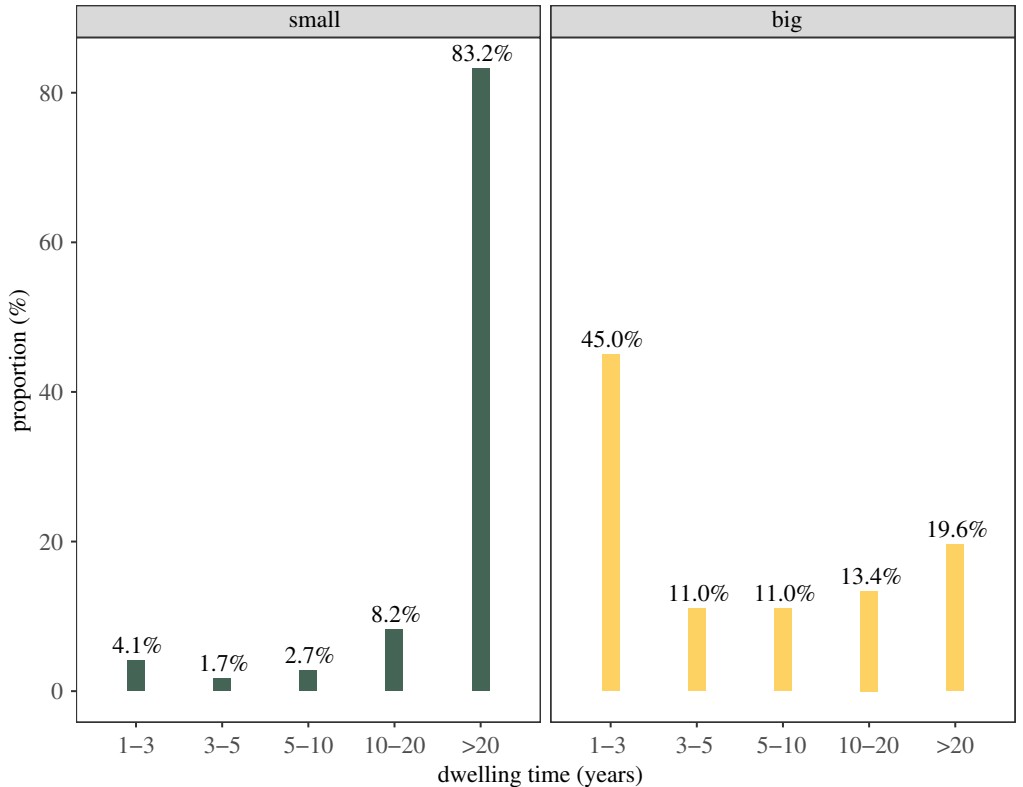

**Figure 2.** The range of dwelling time by community size. Proportion is calculated by the total number of participants among all the small and large communities separately. Almost all the residents' dwelling time exceeds 20 years in small communities, while nearly half of dwelling time in large communities is 1–3 years.

material, table S4 providing demographic differences as well as averages for our key variables between large and small communities).

Table 1 shows the results of principal component analyses on the relevant religiosity items, which yields three domains, 'Religious practice' for a combined measure of religious practice and behaviour, 'Divine punishment/reward' for the degree of personal belief in divine reward for good acts or divine punishment for bad behaviour, and 'Other supernatural belief' for the belief in numerous other superstitions.

## 3.1. Free donation game

Our data follow a highly zero-inflated distribution for donation choice; more than half of interviewees donate all the money to the institution they chose (figure 3). Donations from those in small communities are significantly larger than from those in large communities (Wilcoxon rank sum test: $W = 139880$, $n = 1002$, $p < 0.001$). A $\chi^2$-test also reveals the relationship between community size and a binary variable coding for whether participants donate all the money or not ($\chi^2$ (1, $N = 1002$) $= 19.8$, $p < 0.001$), with those in smaller communities donating all 13.8% more than those in large communities. People with a higher degree of participation in public religious practice are more likely to donate all the money (Wilcoxon rank sum test: $W = 92859$, $n = 1002$, $p < 0.001$).

We find the different religious groups and atheists do have different levels of belief in divine punishment (see electronic supplementary material, table S8). Using the religious denomination as an explanatory variable, we conduct a GLMM regression analysis which shows no association between religious denomination and the free donation amount (see electronic supplementary material, table S9). So those following more punitive religions are not donating more, which is consistent with the results from models in which belief in divine punishment is itself included as a predictor (see below).

We consider a set of candidate models explaining whether participants donate all the endowment to the institution or not by comparing different combinations of the 'community size', 'religiosity' and institution 'choice' variables. The model that contains all these variables has the greatest power for explaining the free donation game results (table 2).

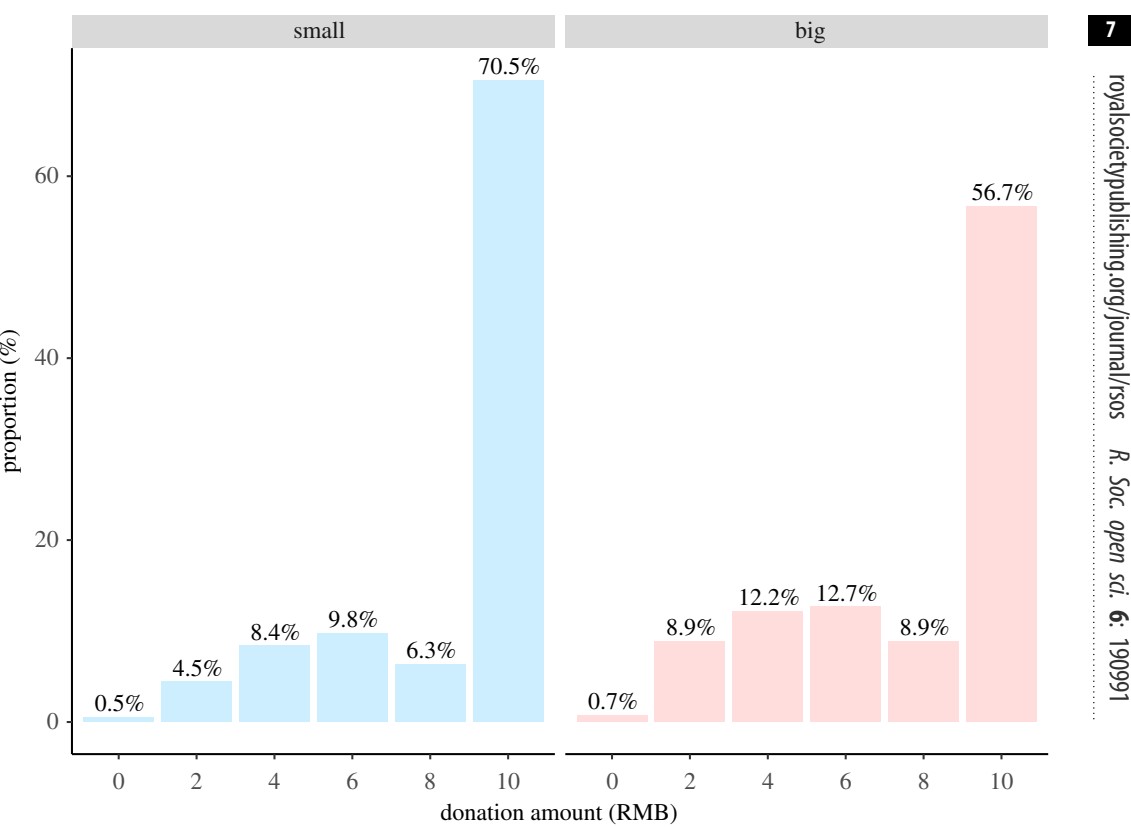

**Figure 3.** Data distribution of donations to the institutions in free donation game. Bars show 70.5% ($N = 292$) of the participants donate all the money to the institutions in small communities, 56.7% ($N = 209$) of participants donate all the money in large communities.

**Table 2.** Model selection results showing the importance of the community size and religiosity on decision-making in free donation game: GLMM is used to predict if all the money is allocated to the institution. Columns report the number of parameters ($K$), the AIC, differences in AIC relative to the minimum in the set ($\Delta$AIC), Akaike weights (AICWt) and the log-likelihood of each model (LL) [56]. Control variables include onlookers, gender, age, offspring, education, economic instability. CS refers to community size, R to three components of religiosity and C to institution choice.

| models | $K$ | AIC | $\Delta$AIC | AICWt | LL |
|---|---|---|---|---|---|
| CS + R + C | 13 | 1126.16 | 0.00 | 0.57 | −550.08 |
| R + C | 12 | 1128.25 | 2.09 | 0.20 | −552.13 |
| CS | 9 | 1128.96 | 2.80 | 0.14 | −555.48 |
| control | 8 | 1129.85 | 3.69 | 0.09 | −556.92 |
| null | 2 | 1164.92 | 38.76 | 0.00 | −580.46 |

Table 3 and figure 4 show that a composite measure of overall religiosity is a significant positive predictor for the amount donated to institutions; when analysed as three principal components, religious practice is a positive predictor, whereas belief in divine punishment and reward is not a significant predictor, which does not support H2. Consistent with H1, participants are more likely to give all in small societies than in large societies, and the choice of a religious institution is not associated with donation size. Poverty (measured here by economic instability, see electronic supplementary material, Methods) reduces the size of donations and education increases them.

Results are robust when using the donation magnitude of those who do not donate the full stake as dependent variable (see electronic supplementary material, table S7). Living in large communities and conducting more religious practice are reliably associated with higher donation amounts, but belief in divine punishment and reward has no overall effect.

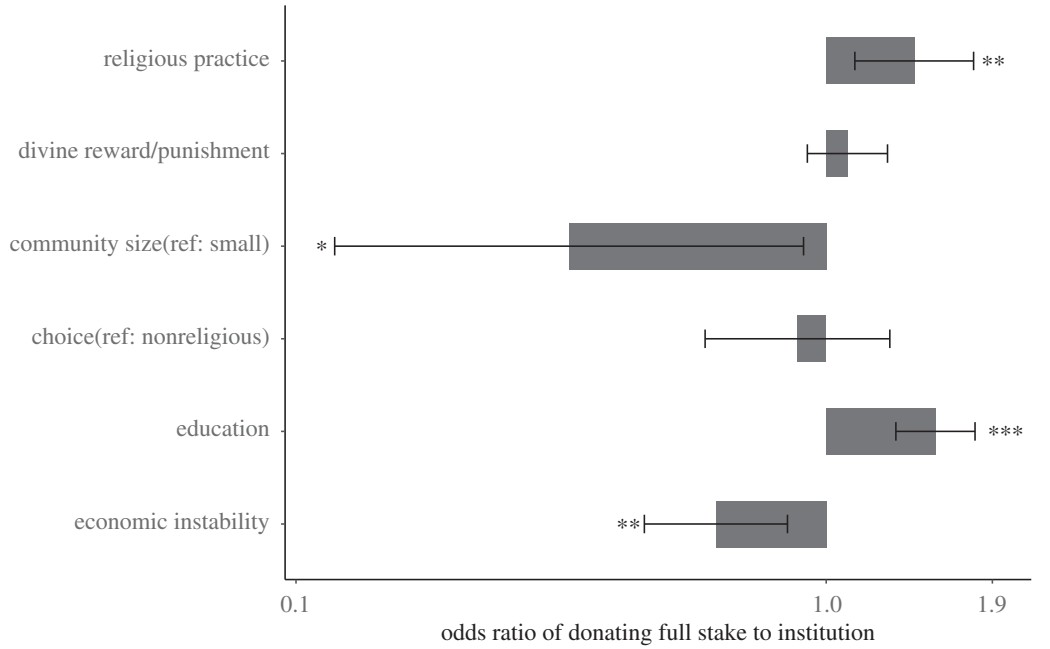

**Figure 4.** Odds ratios with 95% confidence interval plots for coefficients of key variables from the best-selected model on the odds that all the endowment is donated to their recipient institution. Odds ratios greater than 1 indicate an increase and odds ratios less than 1 indicate a decrease in the odds of donating all to their recipient institution (***$p \leq 0.001$, **$p \leq 0.01$, *$p \leq 0.05$). The x-axis is on a logarithmic scale. The model includes other controls ($N = 1002$). Parameter estimates of the best- selected model can be found in electronic supplementary material, table S6.

**Table 3.** Parameter estimates in control and averaged model of determinants of whether donating all the endowment to institution. Survey site is a random effect. Significant effects are in italics.

| | free donation game ($N = 1002$) | | | |
|---|---|---|---|---|
| | control model | | averaging model | |
| | est | s.e. | est | s.e. |
| intercept | −0.506 | 0.495 | −0.665 | 0.48 |
| onlookers | −0.043 | 0.036 | −0.041 | 0.036 |
| gender (ref: man) | 0.181 | 0.169 | 0.232 | 0.173 |
| age | 0.003 | 0.006 | 0.003 | 0.006 |
| offspring | 0.04 | 0.084 | 0.027 | 0.085 |
| education | *0.380* | *0.076* | *0.410* | *0.079* |
| economic instability | *−0.430* | *0.141* | *−0.428* | *0.141* |
| community size (ref: small) | | | *−0.972* | *0.474* |
| choice (ref: non-religious) | | | −0.111 | 0.182 |
| religious practice | | | *0.335* | *0.118* |
| divine punishment/reward | | | 0.084 | 0.079 |
| other supernatural belief | | | 0.133 | 0.081 |

## 3.2. Dice allocation game

We test the overall sample to examine how honest participants appear to have been when reporting the results of their dice throws. The true level of cheating cannot be known, but comparison of the aggregate results with the expected probability distribution can suggest where it may have occurred. The majority of donors choosing religious institutions are religious people, not atheists (see electronic supplementary

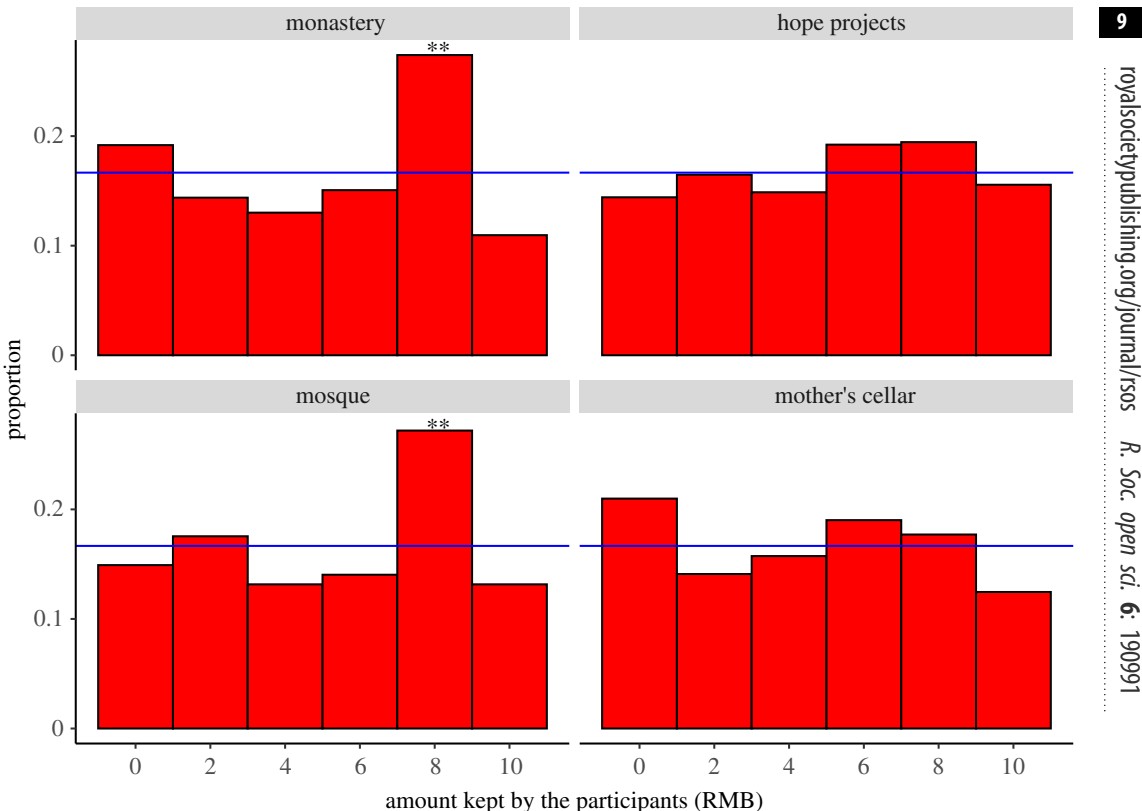

**Figure 5.** The distribution of money kept by the participants by choice of institution to donate to. Blue line represents the theoretical expected frequency ($N = 1002$). Of the four institutions, participants choose two to allocate money to. $^{**}p \leq 0.01$.

material, table S3). We find that participants who donate to a Buddhist monastery are more likely to claim four more points (8 RMB) than expected (Binomial test, $p = 0.005$), and also those who donate to a Mosque have higher than expected claims on four points (Binomial test, $p = 0.004$) (figure 5). However, there is no significant difference between the distributions of reported roll outcomes for different chosen institutions (Kruskal test, $p = 0.82$). Reporting the first or the second dice roll does not appear to show any of the 'bending rules' like that found in Gächter and Shulz's results (where the higher rather than the correct one of the two roles is reported, see electronic supplementary material, figure S2). Bending of the game rules may be changed when behaviour involves interacting with the experimenter, rather than a computer.

At the community level, no other models explain the mean amount kept by the participants between different communities better than the control model (see electronic supplementary material, tables S11 and S12). Contrary to H3 and in support of H4, the effects of community-level religiosity and community size are not significant in this dice game. We also examine whether community size and religiosity are associated with another three measures of possible dishonesty separately (after Gächter and Shulz for no claim, second highest claim and high claim, see electronic supplementary material, table S13). We find stronger belief in divine punishment/reward is not a significant predictor of any measure of community-level dishonesty. Community size and public religious practice are not expected to be important predictors in a game eliminating reputational concerns if donations are motivated by that.

## 4. Discussion

In a free donation game, we find that those in small stable communities (villages) give higher donations than those in large transient communities (cities). More religious practice by individuals is associated with higher donations in this free donation game, whereas fear of divine punishment is not, suggesting that reputational effects rather than supernatural monitoring are motivating donations. This could be because collective ritual or religious activities increase the potential for meeting and gossip [60], which may arouse more concern for reputation. This echoes ethnographic work in some

Indian villages that has shown costlier public religious practice correlate with an improved prosocial reputation [26]. Meanwhile, reputational benefits are accrued from enhanced ability to draw on help [51]. We also find the majority of community members donate the entire endowment to the institutions in our free donation game, which implies that people may cooperate in an uncalculating decision-making to signal their generosity for reputation concerns, behaving in a calculating way is generally perceived as a sign of doubt or uncertainty [61–63]. Generosity as cues of willingness to confer benefit can increase individuals' biological market value, which underlies how to choose partners [64,65]. Some religions are more punitive than others, but religionists belonging to the more punitive religions are not related to more free donation game giving (electronic supplementary material, tables S8 and S9), consistent with the view that secular reputation, as a promoter of cooperation in large-scale societies, outweighs spiritual monitoring. Furthermore, some previous studies revealed substantial gender differences in donation behaviour, that women were significantly more generous than men [66–69], but alike some other literatures [70,71], here we have not found any evidence for gender difference in generosity.

In the dice allocation game, belief in divine punishment/reward does not appear to influence dishonesty. Dishonesty in various game experiments is correlated with cheating behaviours in real life [72], so our results suggest belief in divine punishment/reward may not be an effective mechanism to foster wide-ranging cooperation in real-world settings. Participants' claims also do not vary with size of communities and are uncorrelated with religious practice in this game where reputational concern has been eliminated. More participants who chose religious institutions (which is a public choice) cheat by over-reporting the number four on the dice (figure 5). Players do not appear to want to report 5, probably as it might look like cheating and appear greedy (some may even have cheated in favour of the recipient institution given the low reporting of 5s generally). Likewise, in contrast to findings that females, on average, behave more honestly in cheating games [73,74], our results show there is no effect of gender on dishonesty. Note here we consider sex ratio among communities and whole community level dishonesty in our dice allocation game, not of individual level (electronic supplementary material, table S12).

Findings from both games are consistent with the view that people are signalling their cooperativeness toward religious and non-religious institutions. Reputational concern rather than fear of supernatural monitoring appears to be a more effective mechanism to promote large-scale prosocial tendencies.

The poor are more likely to follow religious norms and have stronger religious belief [75], although in our free donation game poverty appears to reduce the inclination to make larger donations to either secular or religious institutions. In our dice allocation game, we find a strong correlation between average economic instability of participants in a community and the proportion of those people who choose (publicly) to support religious institutions in the game (see electronic supplementary material, figure S3). A possible explanation is that people in these communities are more likely to need help from their peers in the community, and religious individuals are often rated as more trustworthy [76,77]. Religious institutions themselves can also offer direct benefits such as charity or loans.

Overall, day-to-day decisions may thus be focused on securing future help from the community rather than motivated by fear of divine punishment. Although theoretical modelling and laboratory experiments demonstrate the role of sanctioning institutions in the transition to large-scale societies [78,79] and costly punishment in addressing common-pool resource or collective actions dilemmas [79–83], our findings chiming with other works suggest that benefits to cooperation rather than costs from punishment are more likely to be important motivators of cooperation in other real-world contexts or laboratory experiments [60,84–88].

Ethics. The experimental procedures were approved by Lanzhou University (Department of life sciences), and the UCL Research Ethics committee (no. 8669/002)

Data accessibility. The data associated with this research are available within the Dryad repository: https://doi.org/10. 5061/dryad.g48v3m0 [89]. The R code for processing and analyzing the data is available within GitHub: https:// github.com/erhaoGe/RSOS. Auxiliary and confirmatory analyses, as well as demographic information of communities, have been uploaded as supplementary material.

Authors' contributions. Design was done by R.M. and J.-J.W.; hypothesis by R.M.; data collection by E.-H.G. and Y.C.; data analysis by E.-H.G. and Y.C.; writing by E.-H.G., R.M., Y.C. and J.-J.W.

Competing interests. We declare we have no competing interests.

Funding. E.-H.G., Y.C., J.-J.W. and R.M were all funded by Lanzhou University.

Acknowledgements. We thank the many local people who participated in this research, with special thanks to Shiyi Tang, Pengpeng Bai, Liqiong Zhou, Hanzhi Zhang and Sarah Peacey for the great help and assistance they provided.

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
