## [Reviewer comments · Royal Society Open Science]

Review History

RSOS-190991.R0 (Original submission)

Review form: Reviewer 1

Is the manuscript scientifically sound in its present form?

Yes

Are the interpretations and conclusions justified by the results?

Yes

Is the language acceptable?

No

Do you have any ethical concerns with this paper?

No

Have you any concerns about statistical analyses in this paper?

No

Recommendation?

Major revision is needed (please make suggestions in comments)

Comments to the Author(s)

In this manuscript, the authors report a large experiment (N=501) exploring the effect of community size and religiosity on donations and cheating. The results show that donations to both religious and non-religious institutions are motivated by reputational concerns. The same result does not hold in the domain of cheating. Furthermore, neither donations nor cheating are driven by fear of divine punishment.

Overall, the paper is interesting, and I think that a revised version of it can be published. Below is my list of comments, in order of appearance.

Introduction

- There are some recent experiments looking at the effect of group size on cooperation that I think should be mentioned (<https://www.nature.com/articles/srep07937>; <https://journals.plos.org/plosone/article?id=10.1371/journal.pone.0131419>; <https://www.nature.com/articles/s41598-019-41988-3>). These experiments are particularly relevant because, as in the current manuscript, they look at the effect of group size on cooperation in the aseptic setting of a laboratory experiment, rather than in the field. There are also studies looking at group size effect on cooperation using computer simulations, which might be mentioned (<https://journals.aps.org/pre/abstract/10.1103/PhysRevE.84.047102>).
- I am pretty sure that there is quite a lot of literature looking at the correlation between religiosity and pro sociality. This literature should be reviewed.
- The presentation would be clearer if the authors spell out their hypotheses. They could call them H1, H2, etc.

Participants

- The fact that participants were selected at random in public areas does not imply that the sample is representative. For example, if one selects people during the morning, then one does not have represented people who usually go out in the evening and nights. And so on. Plus, it is really unlikely that people choose other people really at random. This is not a big issue, though, just eliminate any reference to representative samples.

Method

- The description of the method is very confusing: the terminology Experiment 1 and Experiment 2 makes readers think that these are actually two different experiments. They are not. They are actually the same experiments with two stages. They should be call Stage 1 and Stage 2.
- Please include more details about the measures of religiosity. Since these are important measures, detailed should be included in the main text and not in the SM.

Results

- Since recent meta-analyses show that women donate more than men (<https://www.ncbi.nlm.nih.gov/pubmed/26913619>; <https://www.sciencedirect.com/science/article/pii/S0165176518301952>), I think that the authors should discuss gender differences in donations in more details, and relate this with earlier research
- Since recent meta-analyses show that women are more honest than men (<http://journal.sjdm.org/18/18619a/jdm18619a.html>; <https://psycnet.apa.org/record/2018-66786-001>), I think that the authors should discuss gender differences in cheating in more details, and relate this with earlier research
- The analysis would be more convincing if the authors add an interaction term to look at

potential differences in the donation game vs the cheating game. It is not indeed very clear that some results (e.g., the relation between community size and donation/cheating) hold in one cases but not in the other one.

Discussion

- "We also detected the majority donated all the money to the institutions in our free donation experiment, in contrast to findings that people donate between half to nothing in some other public goods game experiments or dictator games [12,46,47]". This is not very surprising: donating to an institution is very different from donating to another person in the dictator game.

- "cooperate in an uncalculating decision-making to signal their generosity for reputation concerns, behaving in a calculating way is generally perceived as a sign of doubt or uncertainty [48,49]". Another paper in support of the fact that uncalculating decision-making is a signal of altruism: <https://www.sciencedirect.com/science/article/pii/S2214804316300878>

General comment

- Please revise your English. I've noticed several typos and mistakes.

Review form: Reviewer 2

Is the manuscript scientifically sound in its present form?

No

Are the interpretations and conclusions justified by the results?

Yes

Is the language acceptable?

No

Do you have any ethical concerns with this paper?

No

Have you any concerns about statistical analyses in this paper?

No

Recommendation?

Major revision is needed (please make suggestions in comments)

Comments to the Author(s)

In this manuscript, the authors conduct two donation experiments (free donation game and dice allocation game) on a large naturalistic sample of 501 people in 17 communities, with varying religions or none, ranging from small villages to large cities. The authors want to know whether people in small, stable communities are more cooperative than people in large, less stable communities. They make four predictions and test them. The results show that donations to both religious and non-religious institutions are being motivated by reputational considerations. Besides, fear of divine punishment is not the more salient motive for cooperative behavior. Overall, this is an interesting study. However, there are some remaining issues with the manuscript, requiring some answers.

Major issues:

1) First, I suggest the authors explain why the number of rounds of two games is set to 2. Are the

environments of these two rounds the same? Are onlookers consistent?

- 2) The authors claim that reputational considerations can be more salient in small communities where the people have the opportunity to get to know each other. But in my opinion, it's not just reputation that works here. There are also other factors, such as kin selection. I believe that excluding the influence of other factors will lead to a better understanding of the present results.
- 3) In the experimental design stage for free donation game, the whole game process is observable to onlookers. I have some questions. Is the onlooker a game participant or a passer-by? Are these onlookers familiar with the game player? Are game designers included?
- 4) For Dice allocation game, the authors design that the dices were unobservable by anyone except the participant. Thus I want to know how the authors record the number of the participants who cheated in the game.
- 5) How can the authors draw the following conclusion: People with higher self-reported religiosity were more likely to donate all the money (line 12 of page 16)? What does self-reported religiosity mean? I didn't find it again in the article.
- 6) In figure 3, the authors only compare the proportion of donating all in smaller communities and large communities, and claim that those in smaller communities donating all 13.8% are more than those in large communities. However, for other donations (0, 2, 4, 6, 8), large communities have more advantages than smaller communities. I think that it is better for authors to describe these results and explain them.
- 7) In the manuscript, the authors mention divine punishment. It would be interesting to discuss the relationship between divine punishment and the costly punishment (e.g., New J. Phys. 16 (2014) 083016, Phys. Rev. E 92 (2015) 012819, and PLoS Comput. Biol. 14 (2018) e1006347).

Minor issues:

- 1) In lines 37&38 of page 5, there should be a space between "[13]," and "but".
- 2) In lines 14&15 of page 8, there should be a space between "1)." and "This".
- 3) In lines 12&13 of page 12, "All the dice were..." should be changed to "All the dices were...".
- 4) In lines 22&23 of page 13, "Akaike Information Criterion (AICc)" should be changed to "Akaike Information Criterion (AIC)".
- 5) In lines 33&34 of page 13, "in R3.4.1[42],including ..." should be changed to "in R3.4.1[42], including ...".
- 6) In Table 2 "n=1002", while in Figure 5 "N=1002". Are "n" and "N" are two different variables?
- 7) In lines 3&4 of page 24, "individuals' biological market value, which underlie how to choose partners[50,51]." should be changed to "individuals' biological market value, which underlies how to choose partners [50, 51].".
- 8) In line 51 of page 24, "Reputational concern rather than fear of supernatural monitoring, appear to be a more..." should be changed to "Reputational concern rather than fear of supernatural monitoring, appears to be a more...".
- 9) In lines 27&28 of page 25, "Our findings chimes with other work..." should be changed to "Our findings chime with other work...".
- 10) Finally, the format of references needs careful revision. For example, refs. 1, 13, 14, 17, 19, 22.

Decision letter (RSOS-190991.R0)

19-Jun-2019

Dear Dr Ge,

The editors assigned to your paper ("Large-scale cooperation driven by reputation, not fear of divine punishment") have now received comments from reviewers. We would like you to revise

your paper in accordance with the referee and Associate Editor suggestions which can be found below (not including confidential reports to the Editor). Please note this decision does not guarantee eventual acceptance.

Please submit a copy of your revised paper before 12-Jul-2019. Please note that the revision deadline will expire at 00.00am on this date. If we do not hear from you within this time then it will be assumed that the paper has been withdrawn. In exceptional circumstances, extensions may be possible if agreed with the Editorial Office in advance. We do not allow multiple rounds of revision so we urge you to make every effort to fully address all of the comments at this stage. If deemed necessary by the Editors, your manuscript will be sent back to one or more of the original reviewers for assessment. If the original reviewers are not available, we may invite new reviewers.

- Data accessibility

<http://datadryad.org/submit?journalID=RSOS&manu=RSOS-190991>

- Competing interests

- Authors' contributions

- Acknowledgements

- Funding statement

on behalf of Dr Matjaz Perc (Associate Editor) and Kevin Padian (Subject Editor)
openscience@royalsociety.org

Comments to Author:

Reviewers' Comments to Author:

Reviewer: 1

Comments to the Author(s)

In this manuscript, the authors report a large experiment (N=501) exploring the effect of community size and religiosity on donations and cheating. The results show that donations to both religious and non-religious institutions are motivated by reputational concerns. The same result does not hold in the domain of cheating. Furthermore, neither donations nor cheating are driven by fear of divine punishment.

Overall, the paper is interesting, and I think that a revised version of it can be published. Below is my list of comments, in order of appearance.

Introduction

- There are some recent experiments looking at the effect of group size on cooperation that I think should be mentioned (<https://www.nature.com/articles/srep07937>; <https://journals.plos.org/plosone/article?id=10.1371/journal.pone.0131419>; <https://www.nature.com/articles/s41598-019-41988-3>). These experiments are particularly relevant because, as in the current manuscript, they look at the effect of group size on cooperation in the aseptic setting of a laboratory experiment, rather than in the field. There are also studies looking at group size effect on cooperation using computer simulations, which might be mentioned (<https://journals.aps.org/pre/abstract/10.1103/PhysRevE.84.047102>).
- I am pretty sure that there is quite a lot of literature looking at the correlation between religiosity and pro sociality. This literature should be reviewed.
- The presentation would be clearer if the authors spell out their hypotheses. They could call them H1, H2, etc.

Participants

- The fact that participants were selected at random in public areas does not imply that the sample is representative. For example, if one selects people during the morning, then one does not have represented people who usually go out in the evening and nights. And so on. Plus, it is really unlikely that people choose other people really at random. This is not a big issue, though, just eliminate any reference to representative samples.

Method

- The description of the method is very confusing: the terminology Experiment 1 and Experiment 2 makes readers think that these are actually two different experiments. They are not. They are actually the same experiments with two stages. They should be call Stage 1 and Stage 2.
- Please include more details about the measures of religiosity. Since these are important measures, detailed should be included in the main text and not in the SM.

Results

- Since recent meta-analyses show that women donate more than men (<https://www.ncbi.nlm.nih.gov/pubmed/26913619>; <https://www.sciencedirect.com/science/article/pii/S0165176518301952>), I think that the authors should discuss gender differences in donations in more details, and relate this with earlier research
- Since recent meta-analyses show that women are more honest than men (<http://journal.sjdm.org/18/18619a/jdm18619a.html>; <https://psycnet.apa.org/record/2018-66786-001>), I think that the authors should discuss gender differences in cheating in more details, and relate this with earlier research
- The analysis would be more convincing if the authors add an interaction term to look at potential differences in the donation game vs the cheating game. It is not indeed very clear that some results (e.g., the relation between community size and donation/cheating) hold in one cases but not in the other one.

Discussion

- "We also detected the majority donated all the money to the institutions in our free donation experiment, in contrast to findings that people donate between half to nothing in some other public goods game experiments or dictator games [12,46,47]". This is not very surprising: donating to an institution is very different from donating to another person in the dictator game.

- “cooperate in an uncalculating decision-making to signal their generosity for reputation concerns, behaving in a calculating way is generally perceived as a sign of doubt or uncertainty [48,49]”. Another paper in support of the fact that uncalculating decision-making is a signal of altruism: <https://www.sciencedirect.com/science/article/pii/S2214804316300878>

General comment

- Please revise your English. I’ve noticed several typos and mistakes.

Reviewer: 2

Comments to the Author(s)

In this manuscript, the authors conduct two donation experiments (free donation game and dice allocation game) on a large naturalistic sample of 501 people in 17 communities, with varying religions or none, ranging from small villages to large cities. The authors want to know whether people in small, stable communities are more cooperative than people in large, less stable communities. They make four predictions and test them. The results show that donations to both religious and non-religious institutions are being motivated by reputational considerations. Besides, fear of divine punishment is not the more salient motive for cooperative behavior. Overall, this is an interesting study. However, there are some remaining issues with the manuscript, requiring some answers.

Major issues:

- 1) First, I suggest the authors explain why the number of rounds of two games is set to 2. Are the environments of these two rounds the same? Are onlookers consistent?
- 2) The authors claim that reputational considerations can be more salient in small communities where the people have the opportunity to get to know each other. But in my opinion, it's not just reputation that works here. There are also other factors, such as kin selection. I believe that excluding the influence of other factors will lead to a better understanding of the present results.
- 3) In the experimental design stage for free donation game, the whole game process is observable to onlookers. I have some questions. Is the onlooker a game participant or a passer-by? Are these onlookers familiar with the game player? Are game designers included?
- 4) For Dice allocation game, the authors design that the dices were unobservable by anyone except the participant. Thus I want to know how the authors record the number of the participants who cheated in the game.
- 5) How can the authors draw the following conclusion: People with higher self-reported religiosity were more likely to donate all the money (line 12 of page 16)? What does self-reported religiosity mean? I didn't find it again in the article.
- 6) In figure 3, the authors only compare the proportion of donating all in smaller communities and large communities, and claim that those in smaller communities donating all 13.8% are more than those in large communities. However, for other donations (0, 2, 4, 6, 8), large communities have more advantages than smaller communities. I think that it is better for authors to describe these results and explain them.
- 7) In the manuscript, the authors mention divine punishment. It would be interesting to discuss the relationship between divine punishment and the costly punishment (e.g., *New J. Phys.* 16 (2014) 083016, *Phys. Rev. E* 92 (2015) 012819, and *PLoS Comput. Biol.* 14 (2018) e1006347).

Minor issues:

- 1) In lines 37&38 of page 5, there should be a space between “[13],” and “but”.
- 2) In lines 14&15 of page 8, there should be a space between “1.” and “This”.
- 3) In lines 12&13 of page 12, “All the dice were...” should be changed to “All the dices were...”.
- 4) In lines 22&23 of page 13, “Akaike Information Criterion (AICc)” should be changed to “Akaike Information Criterion (AIC)”.

- 5) In lines 33&34 of page 13, "in R3.4.1[42],including ..." should be changed to "in R3.4.1[42], including ...".
- 6) In Table 2 "n=1002", while in Figure 5 "N=1002". Are "n" and "N" are two different variables?
- 7) In lines 3&4 of page 24, "individuals' biological market value, which underlie how to choose partners[50,51]." should be changed to "individuals' biological market value, which underlies how to choose partners [50, 51].".
- 8) In line 51 of page 24, "Reputational concern rather than fear of supernatural monitoring, appear to be a more..." should be changed to "Reputational concern rather than fear of supernatural monitoring, appears to be a more...".
- 9) In lines 27&28 of page 25, "Our findings chimes with other work..." should be changed to "Our findings chime with other work...".
- 10) Finally, the format of references needs careful revision. For example, refs. 1, 13, 14, 17, 19, 22.

Author's Response to Decision Letter for (RSOS-190991.R0)

See Appendix A.

RSOS-190991.R1 (Revision)

Review form: Reviewer 1

Is the manuscript scientifically sound in its present form?

Yes

Are the interpretations and conclusions justified by the results?

Yes

Is the language acceptable?

Yes

Do you have any ethical concerns with this paper?

No

Have you any concerns about statistical analyses in this paper?

No

Recommendation?

Accept as is

Comments to the Author(s)

Thanks for addressing all my comments.

Review form: Reviewer 2

Is the manuscript scientifically sound in its present form?

Yes

Are the interpretations and conclusions justified by the results?

Yes

Is the language acceptable?

Yes

Do you have any ethical concerns with this paper?

No

Have you any concerns about statistical analyses in this paper?

No

Recommendation?

Accept as is

Comments to the Author(s)

In the revised manuscript, the authors have addressed my comments accordingly, and I would like to recommend the publication of the work in Royal Society Open Science.

Decision letter (RSOS-190991.R1)

29-Jul-2019

Dear Dr Ge,

I am pleased to inform you that your manuscript entitled "Large-scale cooperation driven by reputation, not fear of divine punishment" is now accepted for publication in Royal Society Open Science.

Kind regards,

Andrew Dunn
Royal Society Open Science Editorial Office
Royal Society Open Science
openscience@royalsociety.org

on behalf of Professor Matjaz Perc (Associate Editor) and Kevin Padian (Subject Editor)
openscience@royalsociety.org

Reviewer comments to Author:

Reviewer: 1

Comments to the Author(s)

Thanks for addressing all my comments.

Reviewer: 2

Comments to the Author(s)

In the revised manuscript, the authors have addressed my comments accordingly, and I would like to recommend the publication of the work in Royal Society Open Science.

Appendix A

Answer to reviewer's comments:

Reviewer: 1

In this manuscript, the authors report a large experiment (N=501) exploring the effect of community size and religiosity on donations and cheating. The results show that donations to both religious and non-religious institutions are motivated by reputational concerns. The same result does not hold in the domain of cheating. Furthermore, neither donations nor cheating are driven by fear of divine punishment.

Overall, the paper is interesting, and I think that a revised version of it can be published. Below is my list of comments, in order of appearance.

Introduction

- There are some recent experiments looking at the effect of group size on cooperation that I think should be mentioned (<https://www.nature.com/articles/srep07937>; <https://journals.plos.org/plosone/article?id=10.1371/journal.pone.0131419>; <https://www.nature.com/articles/s41598-019-41988-3>). These experiments are particularly relevant because, as in the current manuscript, they look at the effect of group size on cooperation in the aseptic setting of a laboratory experiment, rather than in the field. There are also studies looking at group size effect on cooperation using computer simulations, which might be mentioned (<https://journals.aps.org/pre/abstract/10.1103/PhysRevE.84.047102>).

Response: *we thank reviewer 1 for helpful comments and suggestions. We have cited the suggested paper.*

- I am pretty sure that there is quite a lot of literature looking at the correlation between religiosity and pro sociality. This literature should be reviewed.

Response: *we have added references in Introduction section : “Multiple facets of religion have emphasized its role as drivers of prosociality [27–31], but empirical findings on the relationship between religiosity and prosocial behaviour are mixed [32–35].”*

- The presentation would be clearer if the authors spell out their hypotheses. They could call them H1, H2, etc.

Response: *we have added H1, H2, H3, H4 in the end of Introduction section and echoed them in our results section.*

Participants

- The fact that participants were selected at random in public areas does not imply that the sample is representative. For example, if one selects people during the morning, then one does not have represented people who usually go out in the evening and nights. And so on. Plus, it is really unlikely that people choose other people really at random. This is not a big issue, though, just eliminate any reference to representative samples.

Response: *we have deleted any words about “representative” in 2.1 Participants section.*

Method

- The description of the method is very confusing: the terminology Experiment 1 and Experiment 2 makes readers think that these are actually two different experiments. They are not. They are actually the same experiments with two stages. They should be call Stage 1 and Stage 2.

Response: *we have changed the terminology Experiment 1 and Experiment 2 to “free donation game” and “dice allocation game” to make it more clear, which means two games of one experiment. We didn’t call them Stage 1 and Stage 2 given we call stage1 as “Choosing institutions” and stage2 as “Experiment” (As shown in Figure 1).*

- Please include more details about the measures of religiosity. Since these are important measures, detailed should be included in the main text and not in the SM.

Response: *we have added the description of the measures of religiosity as “For measuring the religiosity, we asked about the importance of religion..... along with the geographical distance between experiment location and the religious institution they chose” in the end of 2.2 Measures and procedure;*

At the beginning of 2.3 Statistical analysis we added “Variables measuring participants’ religiosity are named as “Importance of religion” “Religious institution distance”;

Moreover, we move our PCA of religiosity to main text as Table 1.

Results

- Since recent meta-analyses show that women donate more than men (<https://www.ncbi.nlm.nih.gov/pubmed/26913619>; <https://www.sciencedirect.com/science/article/pii/S0165176518301952>), I think that the authors should discuss gender differences in donations in more details, and relate this with earlier research

Response: *we have discussed gender differences and added suggested literatures in discussion section as “Furthermore, some previous studies revealed substantial gender differences in donation behaviour..... gender difference in generosity”.*

- Since recent meta-analyses show that women are more honest than men (<http://journal.sjdm.org/18/18619a/jdm18619a.html>; <https://psycnet.apa.org/record/2018-66786-001>), I think that the authors should discuss gender differences in cheating in more details, and relate this with earlier research

Response: *we have discussed gender differences in dice allocation game and added suggested literatures in discussion section as “Likewise, in contrast to findings that females..... not gender factor of individual level”.*

- The analysis would be more convincing if the authors add an interaction term to look at potential differences in the donation game vs the cheating game. It is not indeed very clear that some results (e.g., the relation between community size and donation/cheating) hold in one case but not in the other one.

Response: *I am afraid we could only discuss these two games respectively, for in our dice allocation game, as experimenter we have no information of whether each individual is cheating or not, what we know and have analysis on is the level of honesty of the whole community—comparing the distribution of each dice number of the community to the theoretical distribution, which is, if all participants reported honestly, then each number should appear with an equal probability of 1/6. Thus, the dice allocation game analyses in community level.*

Whereas, the free donation game analyses are in individual level because we focus on their individual division decision. Hence, we could not add an interaction term in the analysis. Thus, we could not address the relation between community size and donation/ cheating.

Discussion

- “We also detected the majority donated all the money to the institutions in our free donation experiment, in contrast to findings that people donate between half to nothing in some other public goods game experiments or dictator games [12,46,47]”. This is not very surprising: donating to an institution is very different from donating to another person in the dictator game.

Response: *we have deleted this sentence.*

- “cooperate in an uncalculating decision-making to signal their generosity for reputation concerns, behaving in a calculating way is generally perceived as a sign of doubt or uncertainty [48,49]”. Another paper in support of the fact that uncalculating decision-making is a signal of altruism: <https://www.sciencedirect.com/science/article/pii/S2214804316300878>

Response: *we have added the ref to this discussion part.*

General comment

- Please revise your English. I’ve noticed several typos and mistakes.

Response: *we have proofread the whole paper and revised these grammatical mistakes.*

Reviewer: 2

In this manuscript, the authors conduct two donation experiments (free donation game and dice allocation game) on a large naturalistic sample of 501 people in 17 communities, with varying religions or none, ranging from small villages to large cities. The authors want to know whether people in small, stable communities are more cooperative than people in large, less stable communities. They make four predictions and test them. The results show that donations to both religious and non-religious institutions are being motivated by reputational considerations. Besides, fear of divine punishment is not the more salient motive for cooperative behavior. Overall, this is an interesting study. However, there are some remaining issues with the manuscript, requiring some answers.

Major issues:

1) First, I suggest the authors explain why the number of rounds of two games is set to 2. Are the environments of these two rounds the same? Are onlookers consistent?

Response: *we asked the players to select two institutions to which they would like to donate money before the gameplay. All participants undertook two games (free donation game and dice allocation game), which were each performed twice, once for each institution they selected. So that there are two rounds of two games.*

Onlookers varied between rounds, apart from which the environments of these two rounds were the same. We have added more details about calculation of onlookers in our 2.2.1 Free donation game as “The number of onlookers.....”

2) The authors claim that reputational considerations can be more salient in small communities where the people have the opportunity to get to know each other. But in my opinion, it's not just reputation that works here. There are also other factors, such as kin selection. I believe that excluding the influence of other factors will lead to a better understanding of the present results.

Response: *we conducted two games, one free donation game in public and one dice allocation in private to see their reputational considerations because in private situation the participants shouldn't care their reputation since no one would know if s/he cheated or not, whilst in public situation they had onlookers around so that reputation could be more important consideration.*

The local institution they donated to could benefit their kin but the effect could be diluted by people who aren't, so that kin selection would not be as important as reputation.

We also referred to some relevant literatures (such as Henrich J et al. 2010

Markets, religion, community size, and the evolution of fairness and punishment. Science 327, 1480–1484), in which community size is also as a key predictor of results of game experiment, other factors, such as kinship, reciprocity, or status differences were not considered in their anonymous games. Here, donation objects of our two games are institutions, which means who will benefit from the money donated by donors depends entirely on institution, so the beneficiary is totally anonymous to donors.

3) In the experimental design stage for free donation game, the whole game process is observable to onlookers. I have some questions. Is the onlooker a game participant or a passer-by? Are these onlookers familiar with the game player? Are game designers included?

Response: *we have added the explanation in 2.2.1 Free donation game as “The number of onlookers, any passerby and subsequent or previous participant who was around and watching, was recorded at the time of the participants were making their donation decisions. We experimenters weren’t included as onlookers”*

We also added the description of onlookers’ familiar with the game player at the beginning of Results section as “ Participants in small communities have lived there for much longer than those in big communities across our study sites, which we assume is greatly increasing the likelihood that onlookers are familiar with or previously interact with participants in small communities.”...

4) For Dice allocation game, the authors design that the dices were unobservable by anyone except the participant. Thus I want to know how the authors record the number of the participants who cheated in the game.

Response: *Participants were asked to roll a dice in a cup, where only they could see the result of the throw, and then report it to the experimenters. Because only participants could see the result of the dice throw, they had the opportunity to cheat, if they chose to do so, without detection and therefore without reputational concerns becoming an issue.*

As experimenter we have no information of whether each individual is cheating or not, what we did is to compare the distribution of each dice number of the community to the theoretical distribution (which is 1/6) to show the level of honesty of the community, the same setting as ref Gächter et al 2016.

5) How can the authors draw the following conclusion: People with higher self-reported religiosity were more likely to donate all the money (line 12 of page 16)? What does self-reported religiosity mean? I didn't find it again in the article.

Response: *we have replaced phrase of “self-reported religiosity” to “degree of participation in public religious practice”, which is one of our component factors (named as “Religious practice”) of religiosity in PCA results.*

6) In figure 3, the authors only compare the proportion of donating all in smaller communities and large communities, and claim that those in smaller communities donating all 13.8% are more than those in large communities. However, for other donations (0, 2, 4, 6, 8), large communities have more advantages than smaller communities. I think that it is better for authors to describe these results and explain them.

Response: *We have added this analysis in supplementary Table 7, which contains a binomial component predicting whether donating their full stake to institution and a count component (here, with a poisson distribution) predicting the magnitude of money donated for those who didn't donate all.*

We explained it in Results section as “Results are robust when using the donation magnitude of those who didn't donate the full stake as dependent variable (see Table S7). Living in large communities and conducting more religious practice are reliably associated with higher donation amounts, but belief in divine punishment and reward has no overall effect”.

7) In the manuscript, the authors mention divine punishment. It would be interesting to discuss the relationship between divine punishment and the costly punishment (e.g., New J. Phys. 16 (2014) 083016, Phys. Rev. E 92 (2015) 012819, and PLoS Comput. Biol. 14 (2018) e1006347).

Response: *we have added these suggested references and discussed more about it in our Discussion section as “Although theoretical modeling and lab experiments demonstrate costly punishment in addressing common-pool resource or collective actions dilemmas [79–83]”.*

Minor issues:

- 1) In lines 37&38 of page 5, there should be a space between “[13],” and “but”.
- 2) In lines 14&15 of page 8, there should be a space between “1).” and “This”.
- 3) In lines 12&13 of page 12, “All the dice were...” should be changed to “All the dices were...”.
- 4) In lines 22&23 of page 13, “Akaike Information Criterion (AICc)” should be changed to “Akaike Information Criterion (AIC)”.
- 5) In lines 33&34 of page 13, “in R3.4.1[42], including ...” should be changed to “in R3.4.1[42], including ...”.
- 6) In Table 2 “n=1002”, while in Figure 5 “N=1002”. Are "n" and "N" are two different variables?
- 7) In lines 3&4 of page 24, “individuals’ biological market value, which underlie

how to choose partners[50,51].” should be changed to “individuals’ biological market value, which underlies how to choose partners [50, 51].”.

8) In line 51 of page 24, “Reputational concern rather than fear of supernatural monitoring, appear to be a more...” should be changed to “Reputational concern rather than fear of supernatural monitoring, appears to be a more...”.

9) In lines 27&28 of page 25, “Our findings chimes with other work...” should be changed to “Our findings chime with other work...”.

10) Finally, the format of references needs careful revision. For example, refs. 1, 13, 14, 17, 19, 22.

Response: *we made all the changes in wording suggested.*

For the 4th issue, we used second-order Akaike Information Criterion to explain the full name of AICc, which is used for our dice allocation game analysis given small sample size (17 community level data).